# Learning without training:
# The implicit dynamics of in-context learning

## Abstract

One of the most striking features of Large Language Models (LLMs) is their ability to learn in-context. Namely at inference time an LLM is able to learn new patterns without any additional weight update when these patterns are presented in the form of examples in the prompt, even if these patterns were not seen during training. The mechanisms through which this can happen are still largely unknown. In this work, we show that the stacking of a self-attention layer with an MLP, allows the transformer block to implicitly modify the weights of the MLP layer according to the context. We argue through theory and experimentation that this simple mechanism may be the reason why LLMs can learn in-context and not only during training. Specifically, we show how a transformer block implicitly transforms a context into a low-rank weight-update of its MLP layer.

## 1 Introduction

Large language models (LLMs), powered by the transformer architecture (Vaswani et al., 2017), have revolutionized modern machine learning, with wide-ranging applications in science, industry, and art (Liu et al., 2023; Dong et al., 2024). Despite this impact, the mechanisms behind their impressive emergent properties (Wei et al., 2022; Bubeck et al., 2023) are still not fully understood. One of the most fascinating and compelling of these properties is the ability of LLMs to perform in-context learning (ICL) wherein the model is able to adapt based on information provided in the input prompt, without any changes or modification to the model's underlying weights. Our work is focused on better understanding the mechanisms which enable this advantageous behavior.

Historically, in machine learning, the ability to extract patterns from data has been understood as a dynamical process in which model weights are updated through an optimization procedure (Goodfellow et al., 2016). However, in the case of ICL, the model weights remain unchanged. Instead, LLMs appear to re-organize or reconfigure their internal representations depending on the prompt and this dynamic adjustment allows them to make predictions that are significantly more accurate. This mysterious and extremely helpful property of LLMs has led researchers to conjecture an implicit form of weight updates taking place at inference time when a prompt is consumed (Garg et al., 2022; von Oswald et al., 2023; Dai et al., 2023; Akyürek et al., 2023; Zhang et al., 2024; Huang et al., 2025). And recent works have even been able to justify theoretically this intuition, showing that simplified transformer blocks, trained on toy set ups of linear regression datasets, perform implicit weight updates corresponding to a sort of gradient descent optimization (von Oswald et al., 2023; Dai et al., 2023; Zhang et al., 2024). Together, these works suggest it is possible to understand ICL as a form of implicit finetuning of the original pretrained model. In this work, we follow this intuition of ICL as imposing implicit weight updates and focus on the contextual information property which we believe is key to understanding the underlying effect of ICL. To this end, we introduce the notion of a contextual block, a generalization of a transformer block. We show that layers with this contextual property, when stacked with standard neural networks, implicitly transform a context into a weight update of the very first layer of the subsequent neural network. Through our analysis we are able to provide an explicit formula for this implicit update to the feedforward layer weights, which surprisingly turns out to be a rank-1 matrix. Interestingly, other works such as Meng et al. (2022) have uncovered that explicit updates with similar rank-1 matrices can modify factual information in a LLM. This suggests that these low-rank matrices may be central to the way LLMs organize and process information at inference time.

Namely, our work demonstrates that contextual blocks, such as self-attention layers combined with a neural network, indeed perform a sort of implicit low-rank finetuning that can be explicitly described as a rank-1 matrix update of the MLP weights computed directly from the relative effect on the context. Our main contributions are as follows:

- We introduce the notion of a contextual block formed by a contextual layer stacked with a neural network, generalizing the key properties of a transformer block which enable in-context learning.

- We show that for contextual blocks the context acts as implicit rank-1 update of the MLP weights, and derive an explicit formula for this implicit weight-update corresponding to the marginal effect of the context on the contextual block.

- Using these implicit updates, we uncover an implicit gradient descent learning dynamics which arises as prompt tokens are consumed during inference.

Note that our implicit weight update formula has two parts in the context of transformer blocks with skip connections (see Theorem B.2): a low-rank weight matrix update and a vector update. The former is reminiscent of the updates founds in factual knowledge editing like in Mitchell et al. (2022) or Meng et al. (2022), while the later has strong similarities to the steering vectors found Ilharco et al. (2022), Hendel et al. (2023), or Todd et al. (2023) for instance. In a way, our work connects steering vectors and low-rank matrix edits to the internal mechanisms of the transformer architecture. From a complementary perspective, our main theorems connect to the recent findings of Chen et al. (2024). They demonstrate that for linear attention, it is theoretically impossible to convert a prompt exactly into implicit weight updates of the attention layer matrices without modifying the model architecture. To achieve exact compression, they must introduce a new set of weights in the form of attention layer biases. In contrast, our work shows that for general transformer blocks, the context can be converted *exactly* into weight updates of the MLP layer without any architectural modifications. While our goal is to uncover the natural mechanics of inference rather than to engineer static context compression, this result highlights a fundamental difference between attention and MLP layers: the latter are naturally predisposed to absorb context as weight updates. The tradeoff, however, is that these exact implicit updates are dynamic and depend on the input query tokens. For a more detailed discussion of related works and how our contributions are related to previous works, see Appendix A.

## 2 CONTEXTUAL BLOCKS

In this section, we abstract some key properties of transformers. In particular, we introduce the notion of contextual layer, which generalizes the self-attention layer of transformer blocks. In this setting a contextual block is the composition of a contextual layer with a standard neural network generalizing the notion of a transformer block. Then we prove our main theorem, which shows that the context for contextual blocks acts as a low-rank fine tuning update of the neural network weights. For the sake of simplicity, we state our results in the case of a neural network without skip-connection. The skip-connection case is similar but more complicated and fully worked out in Appendix B.

We call a *contextual layer*, a network layer $A(\cdot)$ that can take a single vector $x$ alone as input yielding an output $A(x)$; or, optionally, $A$ can take in addition a context $C$ (e.g., a sequence of tokens, an image, etc.) along with the vector $x$, yielding the output.

As a prototypical and guiding example of a contextual layer, consider the self-attention layer of a transformer block, where the context $C$ is an instruction prompt consisting of a sequence of context tokens $C = [c_1, \ldots, c_n]$ and $x$ is the query token from which the LLM will make a prediction. Together $C$ and $x$ create a contextualized input prompt $[C, x] = [c_1, \cdots, c_n, x]$, which is the concatenation of the context tokens and the query token. Note that a transformer maps sequences of a given length to a sequence of the same length. Therefore, we take $A(C, x)$ to be the output of the self-attention layer corresponding to last token $x$[1]. In this way, both $A(C, x)$ and $A(x)$ occupy the same output vector space.

---

[1]The situation our main statements deal with is actually a more general one: Namely, we will want to remove from the context $C$ only a part of it $Y$; in this case the notation $A(C\backslash Y, x)$ or $\Phi(C\backslash Y, x)$ will mean the outputs corresponding to *any token* $x \in C\backslash Y$, and not only the last token in the prompt.

Contextual layers produce contextual vectors computed as the difference

$$\delta A_x(C) := A(C, x) - A(x)$$

between the layer output with and without context for a given input $x$. Motivated by this generalization of the self-attention layer as a contextual layer, we now generalize the notion of a full transformer block to define the notion of a *contextual block*:

**Definition 2.1.** *A contextual block is the composition $\Phi_W = \varphi_W \circ A$ consisting of a contextual layer $A$ as above with a neural network $\varphi_W$; i.e., $\varphi_W(z) = f_\theta(Wz + b)$, where $W$ and $b$ are the weights of an initial fully-connected dense layer and $f_\theta(z)$ is the rest of the neural network parameterized by weights $\theta$.*

In what follows, we show that it is possible to replace the effect of a portion of the context $C$ with a direct modification to the weights $W$. For a context $C$ and a given input $x \in C \backslash Y$, a contextual block $A$ essentially transforms any portion $Y \subset C$ into an implicit update of the initial MLP weights so that $W$ becomes $W + \Delta_x W(Y)$. Furthermore, this $\Delta_x W(Y)$ corresponds to a low-rank weight update of $W$. Interpreted another way, this suggests that contextual layers *load* the subsequent network weights so that the information contained in $Y$ is effectively and efficiently transferred via $\Delta_x W(Y)$.

We make this relationship precise in Theorem 2.2 below. Importantly, the formula derived there is exact so the output of the contextual block with the full context, $\Phi_W(C, x)$, is precisely equivalent to the output with a reduced context and modified weights, $\Phi_{W+\Delta_x W(Y)}(C \backslash Y, x)$. Thus, the low-rank weight update $\Delta_x W(Y)$ perfectly captures the effect of the removed context portion $Y$.

**Theorem 2.2.** *Consider a contextual block $\Phi_W = \varphi_W \circ A$ as above formed by a contextual layer $A$ composed with a neural network $\varphi_W$ whose first fully-connected layer has weight matrix $W$. Given a context $C$ and an input $x \in C \backslash Y$, the effect of some portion $Y \subset C$ of the context on the output of $\Phi_W$ corresponds to a weight update $W + \Delta_x W(Y)$. Namely, if $A(C \backslash Y, x) \neq 0$, then we have that*

$$\Phi_W(C, x) = \Phi_{W+\Delta_x W(Y)}(C \backslash Y, x) \quad where \quad \Delta_x W(Y) = \frac{(W\delta A_x(Y))A(C \backslash Y, x)^T}{\|A(C \backslash Y, x)\|^2}, \quad (1)$$

*where $\delta A_x(Y) := A(C, x) - A(C \backslash Y, x)$ is the context vector associated to $Y$. Furthermore, note that since $W\delta A_x(Y)$ is a column vector and $A(C \backslash Y, x)^T$ is a row vector, $\Delta_x W(Y)$ corresponds to a rank-1 weight update.*

*Proof.* The result follows by direct computation. Let $\varphi_W(z) = f_\theta(Wz + b)$, where $W$ and $b$ are the weights of the first dense layer of $\varphi$ and $f_\theta$ represents the rest of the network. Then, we have by definition that

$$\begin{aligned}
\Phi_{W+\Delta_x W(Y)}(C \backslash Y, x) &= \varphi_{W+\Delta_x W(Y)}\Big(A(C \backslash Y, x)\Big) \\
&= f_\theta\Big((W + \Delta_x W(Y))A(C \backslash Y, x) + b\Big) \\
&= f_\theta\Big(WA(C \backslash Y, x) + \Delta_x W(Y)A(C \backslash Y, x) + b\Big).
\end{aligned}$$

Now, replacing $\Delta_x W(Y)$ by its definition given in Eq. 1 and using that $\frac{z^T}{\|z\|^2}z = 1$, we obtain

$$\begin{aligned}
\Phi_{W+\Delta_x W(Y)}(C \backslash Y, x) &= f_\theta\left(WA(C \backslash Y, x) + \frac{(W\delta A_x(Y))A(C \backslash Y, x)^T}{\|A(C \backslash Y, x)\|^2}A(C \backslash Y, x) + b\right) \\
&= f_\theta\Big(W\big(A(C \backslash Y, x) + \delta A_x(Y)\big) + b\Big).
\end{aligned}$$

Finally, by definition of the context vector we have that $A(C \backslash Y, x) + \delta A_x(Y) = A(C, x)$; and therefore,

$$\Phi_{W+\Delta_x W(Y)}(C \backslash Y, x) = f_\theta\left(WA(C, x) + b\right) = \varphi_W\left(A(C, x)\right) = \Phi_W(C, x)$$

which ends the proof. $\square$

**Remark 2.3.** *Our theorem states that* any *contextual layer produces an implicit weight transfer from the prompt to the first neural network layer, implicitly modifying the behavior of the pretrained neural*

*network. Among all possible contextual layers (e.g., self-attention, RNN, or recurrent layers with local attention as in De et al. (2024)), some may be better at providing useful weight modifications than others. It may be interesting to evaluate the generative power of a contextual-layer in terms of the particular form of the implicit weight updates given by our theorem and the special structure of $A$ given by the contextual layer. In Appendix D, we compare RNN-based vs. attention-based conceptual layers from the point of view of ICL.*

**Remark 2.4.** *We observe that the implicit update $\Delta_x W$ is not unique. Namely for any matrix $M$ such that $MA(C\backslash Y, x) = 0$, the update $\Delta_x W + M$ would work as well. This is a manifestation of overparametrization in deep learning and the fact that a given function can be represented by different configations of a single network. Observe however, that the matrices $\Delta_x W$ are mimimal in the sense that they are rank 1.*

**Remark 2.5.** *Geva et al. (2021) have found that the MLP layer in a transformer block functions as a form of key-value store, where the neuron vectors of the first MLP matrix implement the keys while the neuron vectors of the second MLP matrix are the values. In this perspective, it is interesting to note that the effect of the context seems to act as a transformation of the keys leaving the values unchanged.*

Note that when $Y = C$ is the full context, the theorem above gives a formula to put all the context information into the weight matrix $W$. See Figure 1.

**Corollary 2.5.1.** *In the notation above, the full context $C$ can be transferred to the neural network weights by the following update (provided that $A(x) \neq 0$):*

$$\Phi_W(C, x) = \Phi_{W+\Delta_x W(C)}(x), \quad with \quad \Delta_x W(C) = \frac{(W\delta A_x(C))A(x)^T}{\|A(x)\|^2}, \tag{2}$$

*where $\delta A_x(C) = A(C, x) - A(x)$ is the context vector and $\Delta_x W$ is rank-1, since $W\delta A_x(C)$ is a column vector and $A(x)^T$ is a row vector.*

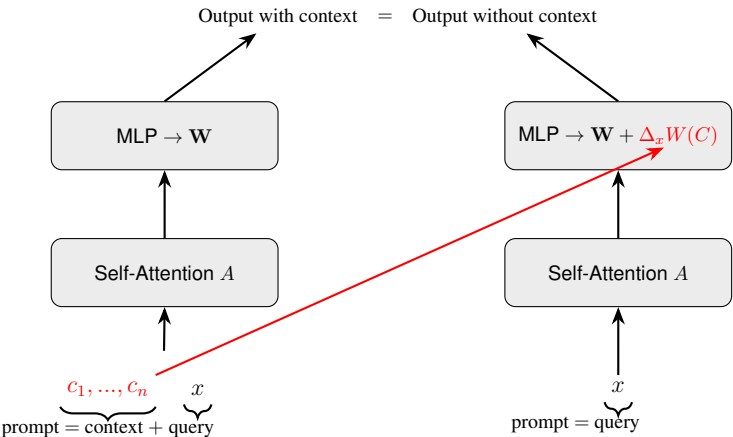

Figure 1: When taking $Y = C$ to be the full context and a query $x$, the corollary to Theorem 2.2 provides an explicit formula which effectively captures how the effect of the context $C$ is encoded as a weight transfer to the first layer MLP weight $W$ via $\Delta_x W(C)$.

**Remark 2.6.** *The weight transfer formula in Eq. 1 can be also rewritten using union/concatenation of context by setting $D = C\backslash Y$; namely:*

$$\Phi_W(D \cup Y, x) = \Phi_{W+\Delta_x W(Y)}(D, x).$$

Another interesting case is when $Y$ corresponds to the user input and $C = [Y, x_1, \ldots, x_n]$, where the $x_i$'s are the generated response tokens. In this case, we can quantify the effect of the user provided context $Y$ on the response generation by a immediate application of Theorem 2.2:

**Corollary 2.6.1.** *In the notation above, we have that*

$$\Phi_W(Y, x_1, \ldots, x_i) = \Phi_{W+\Delta_{x_i} W(Y)}(x_1, \ldots, x_i) \tag{3}$$

*where the implicit update is given by*

$$\Delta_{x_i} W(Y) = \frac{\left(W \delta A_{x_i}(Y)\right) A(x_1, \ldots, x_i)^T}{\|A(x_i, \ldots, x_i)\|^2} \tag{4}$$

*with context vector*

$$\delta A_{x_i}(Y) = A(Y, x_1, \ldots, x_i) - A(x_1, \ldots, x_i). \tag{5}$$

In Appendix B, we generalize Theorem 2.2 for neural networks with skip-connections, as is usually the case for standard transformer blocks. In Appendix C, we explain how to extend this theorem to a stack of transformer blocks by iteratively applying it to each block. In Section 4, we verify our theoretical results experimentally on a standard concrete example.

## 3 THE IMPLICIT LEARNING DYNAMICS OF ICL

With this insight on the relationship between the the context and its implicit affect on the weight parameters, we now use Eq. 2 to examine the weight dynamics of $W$ via in-context learning. Namely, when the context $C = [c_1, \ldots, c_n]$ is a sequence of tokens, an iterative application of Corollary 2.5.1 uncovers an implicit learning dynamics generated by the effect of each context token on the contextual block output.

Starting with the initial weight $W_0$ for $\varphi_W$, the first dense layer of the neural network, we compute the weight updates corresponding to the addition of a new context token $c_i$ provided to us by Corollary 2.5.1. We have

$$
\begin{aligned}
\Phi_{W_0}(c_1, x) &= \Phi_{W_0 + \Delta_x W_0(c_1)}(x) \\
\Phi_{W_0}(c_1, c_2, x) &= \Phi_{W_0 + \Delta_x W_0(c_1, c_2)}(x) \\
&\vdots \\
\Phi_{W_0}(c_1, \ldots, c_n, x) &= \Phi_{W_0 + \Delta_x W_0(c_1, \ldots, c_n)}(x)
\end{aligned}
$$

This leads to the following sequence of corresponding MLP weights

$$
\begin{aligned}
W_1 &= W_0 + \Delta_x W_0(c_1) \tag{6} \\
W_2 &= W_0 + \Delta_x W_0(c_1, c_2) \tag{7} \\
&\vdots \tag{8} \\
W_n &= W_0 + \Delta_x W_0(c_1, \ldots, c_n). \tag{9}
\end{aligned}
$$

By construction, this sequences converges to the effect of the full context on the initial MLP weights so that

$$\Phi_{W_0}(c_1, \ldots, c_n, x) = \Phi_{W_n}(x).$$

The following proposition shows that this implicit learning dynamics is similar to that of online gradient descent, where the tokens play the role of the data points and a loss which changes at each step depending of the token considered for that step.

**Proposition 3.1.** *In the notation above, the iterative process of weight updates can be realized as a form of stochastic gradient updates*

$$W_{i+1} = W_i - h\nabla_W L_i(W_i)$$

*with learning rate given by $h = 1/\|A(x)\|^2$ and loss at step $i$ given by*

$$L_i(W) = \mathrm{trace}(\Delta_i^T W)$$

*where $\Delta_i = W_0\Big(A(c_1, \ldots, c_i, x) - A(c_1, \ldots, c_{i+1}, x)\Big)A(x)^T$.*

*Proof.* Firstly, considering the sequence of $W_i$'s as defined in Eq. 6-9 above and Eq. 2, we have

$$
\begin{aligned}
W_{i+1} - W_i &= \Delta_x W_0(c_1, \ldots, c_{i+1}) - \Delta_x W_0(c_1, \ldots, c_i) \\
&= \frac{W_0\Big(A(c_1, \ldots, c_{i+1}, x) - A(c_1, \ldots, c_i, x)\Big)A(x)^T}{\|A(x)\|^2} \\
&= -h\Delta_i,
\end{aligned}
$$

with $h = 1/\|A(x)\|^2$ and $\Delta_i = W_0\Big(A(c_1, \ldots, c_i, x) - A(c_1, \ldots, c_{i+1}, x)\Big)A(x)^T$.

This means that

$$W_{i+1} = W_i - h\Delta_i = W_i - h\nabla_W \operatorname{trace}(\Delta_i^T W), \qquad (10)$$

since in general we have $\nabla_W \operatorname{trace}(A^T W) = A$. □

Notice that $\Delta_i$ measures the marginal effect of the addition of context token $c_{i+1}$ to the partial context $c_1, \ldots, c_i$. Intuitively, when $c_{i+1}$ has no marginal effect on the output; i.e., when $A(c_1, \ldots, c_i, x) = A(c_1, \ldots, c_{i+1}, x)$, we would expect that the corresponding update to the MLP weights $W$ also vanishes. This intuition is quantitatively justified through Proposition 3.1 since by definition $\Delta_i$ indeed vanishes since $A(c_1, \ldots, c_i, x) - A(c_1, \ldots, c_{i+1}, x)$ is zero. Figure 3 verifies this behavior in a simple experiment showing that these gradients vanish as the learning dynamics converge and the entire context is processed. In short, as the marginal effect of the additional context $c_{i+1}$ goes to zero, so too does the relative change in the MLP weights $W$ and thus their gradient updates.

**Remark 3.2.** *Interestingly, it is possible to derive a different but similar implicit learning dynamics for $W_0, W_1, \ldots, W_n$ by considering partial updates leaving the overall contextual block output unchanged at each step when the partial updates are used in conjunction with the remaining tokens; i.e., define $W_i$ so that $\Phi_{W_i}(c_{i+1}, \cdots, c_n, x) = \Phi_{W_0}(c_1, \ldots, c_n, x)$. These dynamics are described in Appendix E. The difference is that, in general, one can no longer represent the marginal contextual effect by a single gradient update, but instead leads to a factorization formula for the overall weight $W_n$ so that $\Phi_{W_n}(x) = \Phi_{W_0}(c_1, \ldots, c_n, x)$.*

## 4 EXPERIMENTS

In order to verify Theorem 2.2 in practice, we consider a well-defined problem of learning a function class from *in-context* examples. This specific task has been studied throughout the literature (Garg et al., 2022; Akyürek et al., 2023; von Oswald et al., 2023; Ahn et al., 2023; Zhang et al., 2024) and those works are often concerned with the theoretical properties or experimental robustness of ICL to various function classes. In particular, in Zhang et al. (2024) and Garg et al. (2022), the authors show that it is possible to train a transformer from scratch to perform in-context learning of linear functions. That is to say, given a transformer model trained on a class of linear functions, the trained model is able to learn new and unseen linear functions (drawn from a distribution similar to that used during training) purely from in-context examples with performance comparable to the optimal least squares estimator. There the authors were concerned with quantifying how robust transformers are (and are not) to distributional shifts between the training data of the model and inference-time prompts. That is not our goal here.

Instead, since these works have have already verified that transformers can indeed learn linear models in-context, we use a similar experimental framework to verify that the in-context prompts can effectively be transferred to a weight update via Eq. 2. We verify that the prediction made by the trained model with an in-context prompt is identical to the prediction made by the model with MLP weights modified according to Eq. 2 but without access to the in-context prompt.

### 4.1 SETUP

We train a single layer, standard transformer on instances of prompts composed of input-output pairs of the form $(x_1, h(x_1), \ldots, x_N, h(x_N), x_{\text{query}})$ where the $x_i, x_{\text{query}}$ are sampled i.i.d. from a distribution $\mathcal{D}_x$ and the function $h$ is sampled independently from a distribution over functions in a function class $\mathcal{H}$. We take $\mathcal{H}$ to be the class of linear functions defined by $h(x) = \omega^T x$ where each $\omega \sim \mathcal{N}(0, I_d)$ and we sample points $x_i, x_{\text{query}} \sim \mathcal{N}(0, I_d)$. Here the features $x_i, x_{\text{query}}$ are $d$-dimensional and the outputs $y_i$ are scalar. The goal of the in-context learner is to use the input-output pair prompt created from a similarly constructed linear regression task (by sampling a new and unseen $w_{\text{test}}, x_i, x_{\text{query}}$ drawn from the same distributions as used during training) and to form a prediction $\widehat{y}(x_{\text{query}})$ so that $\widehat{y}(x_{\text{query}}) = \omega_{\text{test}}^T x_{\text{query}}$.

Each training prompt is indexed by a task denoted $\tau \in \mathbf{N}$ and we express each prompt as an embedding matrix $E_\tau$ so that

$$E_\tau := \begin{pmatrix} x_{\tau,1} & x_{\tau,2} & \cdots & x_{\tau,N} & x_{\tau,\text{query}} \\ \langle \omega_\tau, x_{\tau,1} \rangle & \langle \omega_\tau, x_{\tau,2} \rangle & \cdots & \langle \omega_\tau, x_{\tau,N} \rangle & 0 \end{pmatrix} \in \mathbb{R}^{(d+1)\times(N+1)}.$$

$$\begin{pmatrix} x_1 & x_2 & \cdots & x_N & x_{\text{query}} \\ f(x_1) & f(x_2) & \cdots & f(x_N) & 0 \end{pmatrix} \longrightarrow f(x_{\text{query}})$$

so that $E_\tau = [C, x]$. Let $\Phi_W = \varphi_W \circ A$ denote our simple transformer consisting of an attention block $A$ followed by a fully connected dense neural neural network $\varphi_W$. Since transformers map sequences of a given length to sequences of the same length, the natural model prediction $\widehat{y}(x_{\tau,\text{query}})$ for $x_{\tau,\text{query}}$ is the last component of the query-token output by a single transformer block[2]; that is,

$$\widehat{y}(x_{\tau,\text{query}}) = \Phi_W(E_\tau)_{(d+1),(N+1)} \tag{11}$$

Note that, defined this way, the dimensionality of $\Phi_W(E_\tau) = \Phi_W([C, x])$ and $\Phi_{W+\Delta_x W}(x)$ agree. We train the transformer using the loss over a batch of size $B$ defined as

$$\widehat{L}(\theta) = \frac{1}{2B} \sum_{\tau=1}^{B} \left( \widehat{y}_{\tau,\text{query}} - w_\tau^T x_{\tau,\text{query}} \right)^2.$$

### 4.1.1 TRANSFORMER SETUP

For our experiments, we focus on autoregressive ("decoder-only") models consisting of a single-layer transformer $\Phi_W = \varphi_W \circ A$ with a multi-head self-attention block $A$ followed by a dense, fully-connected, two-layer MLP $\varphi_W$. Specifically, for an input $X = [x_1, \ldots, x_{N+1}]$ written as a sequence of vectors $x_i \in \mathbb{R}^{d+1}$, we have

$$\begin{aligned} A(X; W_H, W_Q, W_K, W_V) &= \text{MultiHeadAttn}(X; W_H, W_Q, W_K, W_V) \\ &= W_H[H_1, \cdots, H_h]. \end{aligned}$$

Each head $H_i$ is defined as usual by

$$H_i = \text{Attn}(X; W_{Q,i}, W_{K,i}, W_{V,i}) = W_{V,i} \cdot X \cdot \text{softmax} \frac{(W_{K,i}X)^T W_{Q,i}X}{\sqrt{d_k}}$$

with $W_{K,i}, W_{Q,i}, W_{V,i} \in \mathbb{R}^{d_k \times (d+1)}$, $W_H \in \mathbb{R}^{(d+1) \times d_{\text{model}}}$.

The MLP itself is a two-layer ReLU neural network

$$\varphi_W(x) = W' \text{ReLU}(Wx + b) + b',$$

with $b, b' \in \mathbb{R}^{d_{\text{mlp}}}$ and $W \in \mathbb{R}^{d_{\text{mlp}} \times (d+1)}$. For our initial experiments we don't employ MLP skip connections, LayerNorm or any form of positional encoding. In the experiments that follow, we take $d = 2$, $N = 100$, $d_{\text{mlp}} = 128$, $d_{\text{model}} = 32$ and $d_k = d_{\text{model}}/h$ with number of heads $h = 8$.

### 4.2 VERIFYING THEOREM 2.2

Given a transformer trained on linear functions, we show that the in-context prompt can be transferred to a weight update as defined in Eq. 2. Namely we want to show that

$$\Phi_W(C, x) = \Phi_{W+\Delta_x W}(x);$$

or equivalently, for a task $\tau$,

$$\Phi_W \left( \begin{pmatrix} x_{\tau,1} & x_{\tau,2} & \cdots & x_{\tau,N} & x_{\tau,\text{query}} \\ \langle \omega_\tau, x_{\tau,1} \rangle & \langle \omega_\tau, x_{\tau,2} \rangle & \cdots & \langle \omega_\tau, x_{\tau,N} \rangle & 0 \end{pmatrix} \right) = \Phi_{W+\Delta W} \left( \begin{pmatrix} x_{\tau,\text{query}} \\ 0 \end{pmatrix} \right)$$

where

$$\Delta W = \Delta_{x_{\tau,\text{query}}} W \left( \begin{pmatrix} x_{\tau,1} & x_{\tau,2} & \cdots & x_{\tau,N} \\ \langle \omega_\tau, x_{\tau,1} \rangle & \langle \omega_\tau, x_{\tau,2} \rangle & \cdots & \langle \omega_\tau, x_{\tau,N} \rangle \end{pmatrix} \right)$$

is computed as in Eq. 2. Figure 2 compares the validation loss obtained by using each side of the equation above to make predictions upon an evaluation query-token. The loss values for both setups are reported for each checkpoint obtained during pretraining. We can see that these losses are the same for the two computations (left, middle), and this behavior is evidenced over 100 newly sampled tasks for all points $(x_1, x_2) \in \mathbb{R}^{d=2}$ (right).

---

[2]For the sake of simplicity, in Theorem 2.2 and in this experiment, we use standard transformer blocks Vaswani et al. (2017) but without the skip connection on the MLP layer; see Appendix B to learn how to deal with the skip connection.

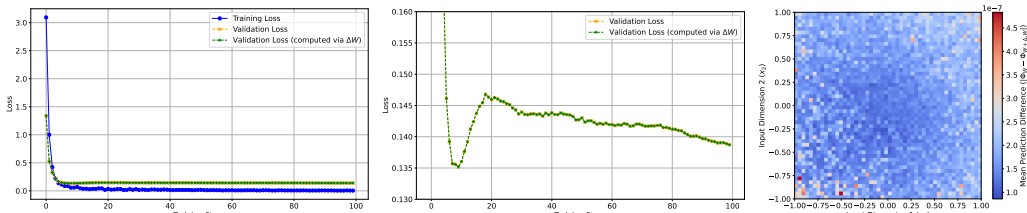

Figure 2: Train and Validation loss curves. Here, the "Validation loss (computed via $\Delta W$)" refers the loss computed using $\Phi_{W+\Delta W}$; i.e., the trained model prediction given only $x_{\text{query}}$ but with MLP weights modified by $\Delta W$ as defined in Eq. 2. **Left:** Training loss and both validation Loss curves. **Middle:** Close-up of validation loss computed both ways; i.e., using $\Phi_W(C, x)$ vs. $\Phi_{W+\Delta_x W}(x)$. **Right:** Once trained, we sample 100 test tasks and for each point $(x_1, x_2) \in \mathbb{R}^{d=2}$ average the difference between $\Phi_W$ and $\Phi_{W+\Delta_x W}$. The two outputs agree on a wide range of both tasks and input values up to an order of $10^{-7}$.

### 4.3 CONVERGENCE OF $\Delta W$

The experiments in this section aim to understand how the weights adapt as the in-context prompt is processed by the model during the implicit learning dynamics described by Proposition 3.1. In particular, we want to verify that the gradient updates vanish as context convergence is reached.

We create a sequence $\{(\Delta W)_i\}_{i=1}^N$ where each $(\Delta W)_i$ is as described in Eqs. 6-9. That is, we have that

$$\Phi_W(C_i, x) = \Phi_{W+(\Delta W)_i}(x)$$

where

$$C_i = [c_1, \ldots, c_i] = \begin{pmatrix} x_{\tau,1} & \cdots & x_{\tau,i} \\ \langle \omega_\tau, x_{\tau,1} \rangle & \cdots & \langle \omega_\tau, x_{\tau,i} \rangle \end{pmatrix} \quad \text{and} \quad x = \begin{pmatrix} x_{\tau,\text{query}} \\ 0 \end{pmatrix}.$$

If we let $W_0$ denote the learned weights of the first dense layer, it follows from Corollary 2.5.1, that for any $i = 1, 2, \ldots, N$,

$$(\Delta W)_i = \frac{(W_0 \delta A_x(C_i)) A(x)^T}{\|A(x)\|^2}, \quad \text{where } \delta A_x(C_i) = A(c_1, \ldots, c_i, x) - A(x).$$

Intuitively, we expect that as the 'in-context learner' processes more of the prompt, the relative change in the $(\Delta W)_i$ should decrease. In Figure 3 we verify that this is indeed the case.

For a given context $C_i = [c_1, \ldots, c_i]$ of length $i$, we plot the marginal change in $(\Delta W)_i$ from incorporating one additional context token $c_{i+1}$ which would yield $(\Delta W)_{i+1}$ for the context $C_{i+1} = [c_1, \ldots, c_i, c_{i+1}]$. We measure this marginal change via the L2-norm; i.e., for each context length $i$ we compute (cf. Proposition 3.1)

$$\|\nabla_W L_i(W)\|_2 = \|(\Delta W)_{i+1} - (\Delta W)_i\|_2.$$

We observe in Figure 3 that the gradient updates decrease and vanish as the implicit learning dynamics progresses toward the full context as we expect from a converging gradient descent dynamics.

### 4.4 COMPARISON WITH FINETUNING

For the experiments in this section, we pretrain a transformer as above (i.e., with a multi-head single layer transformer block without MLP skip-connection or LayerNorm) with examples of the form

$$E_\tau := \begin{pmatrix} x_{\tau,1} & x_{\tau,2} & \cdots & x_{\tau,N} & x_{\tau,\text{query}} \\ \langle \omega_\tau, x_{\tau,1} \rangle & \langle \omega_\tau, x_{\tau,2} \rangle & \cdots & \langle \omega_\tau, x_{\tau,N} \rangle & 0 \end{pmatrix} \in \mathbb{R}^{(d+1) \times (N+1)}.$$

For finetuning we create one new test example by sampling a $\omega_{\text{test}}$ and $x_{\text{test}}$ which the model has not seen during pretraining, though drawn from the same distribution during pretraining. Set

$$\mathcal{D}_{\text{FT}} := \begin{pmatrix} x_1 & \cdots & x_M & x_{\text{test}} \\ \langle \omega_{\text{test}}, x_1 \rangle & \cdots & \langle \omega_{\text{test}}, x_M \rangle & 0 \end{pmatrix}$$

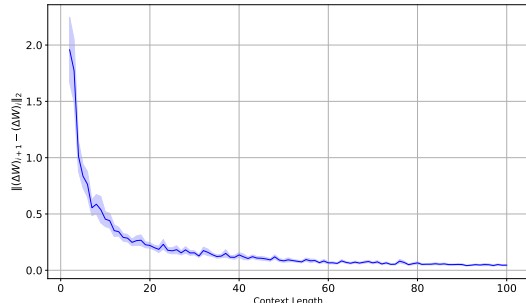

Figure 3: Convergence of $(\Delta W)_i$. As more of the context in processed, the relative change in the weights $W$ converges to zero. For context length $i > 2$, the plot above represents the average difference $\|(\Delta W)_{i+1} - (\Delta W)_i\|_2$ and the standard error over 100 separate trials.

For each $i = 1, 2, \cdots, M$, we create a finetuning dataset consisting of the first $i$ elements of $\mathcal{D}_{\text{FT}}$, ignoring the last column which is our test query. Here we take $M = N$ for consistency but really $M$ could be any value. That is, for all $i = 1, \cdots, M$, set

$$\mathcal{D}_{\text{FT}}^i = \begin{pmatrix} x_1 & x_2 & \cdots & x_i \\ \langle \omega_{\text{test}}, x_1 \rangle & \langle \omega_{\text{test}}, x_2 \rangle & \cdots & \langle \omega_{\text{test}}, x_i \rangle \end{pmatrix}.$$

We initialize the transformer with the pretrained weights, then finetune using SGD taking one example $\begin{pmatrix} x_i \\ \langle \omega_{\text{test}}, x_i \rangle \end{pmatrix}$ at a time in the same order as they are processed in-context, updating the weight matrix $W$ of the MLP layer at each step. After finetuning on all $i$ examples of $\mathcal{D}_{\text{FT}}^i$, we compute the loss of the finetuned model on the test query $(x_{\text{test}}, 0)$. We call this the 'gradient descent (GD) test loss' for $i$ steps.

Similarly, for each $i$ we compute the weight transfer as defined in Eq. 2 using the context

$$C_i = \begin{pmatrix} x_1 & x_2 & \cdots & x_i \\ \langle \omega_{\text{test}}, x_1 \rangle & \langle \omega_{\text{test}}, x_2 \rangle & \cdots & \langle \omega_{\text{test}}, x_i \rangle \end{pmatrix}$$

and the same test query as before $x = (x_{\text{test}}, 0)$. Using the value of $\Delta_x W(C_i)$ from the weight transfer formula, we compute the loss on test query $(x_{\text{test}}, 0)$. We call this the '$\Delta W$ test loss' for context length $i$.

In Figure 4 (left), we compare the finetune SGD test loss with the $\Delta W$ weight transfer test loss showing the average and standard error over 100 separate trials. Although different, we see that the two learning processes (finetuning and implicit weight-update dynamics) minimize the loss in similar ways. Furthermore, we use the (normalized) Frobenius inner product to compare the weight updates to $W$ which arise via finetuning and via the implicit weight update. That is, for each $i \in [M]$, we can compare the weight update to $W$ coming from our implicit dynamics (i.e., $\Delta_x W(C_i)$) and the update coming from finetuning on $\mathcal{D}_{\text{FT}}$, call it $\nabla_W L_i$. As we see in Figure 4 (right), the direction of two weight updates remain highly aligned in weight space as the context length increases and as the number of finetune gradient steps increases.

## 5 CONCLUSION AND FUTURE DIRECTIONS

Our approach to uncovering the transformer's in-context learning mechanics improves upon previous methods in that it does not put any restrictions on the self-attention layer architecture. While earlier theoretical works have also derived a similar form of implicit learning dynamics, these did so only under limiting assumptions on the self-attention layer, such as requiring linear attention or a single head as well as specific forms of prompts; see von Oswald et al. (2023), Dai et al. (2023), and Huang et al. (2025), see also (Shen et al., 2024; Deutch et al., 2024). In contrast our main Theorems (Thm. 2.2 and Thm. B.2) remain valid if the self-attention layer is switched by other forms of contextual layers, such an RNNs, state space models, or any layer that can take an input and optionally a context. This is surprising because our analysis hints that ICL is less about the internals of self-attention, but

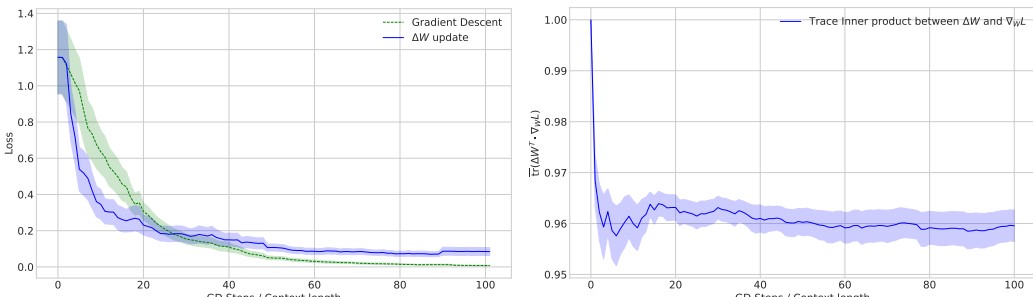

Figure 4: Direct finetuning vs implicit weight update. **Left:** Both finetuning and implicit weight updates minimize the loss in similar ways. **Right:** The two forms of weight updates remain highly aligned with respect to the normalized Frobenius inner product.

rather about the fact that regular neural networks can transfer modification of input space to their weight structure. This is a deep property that has been noticed in a number of theoretical works, and has been used to helped understand why deep neural networks generalize so well (Seong et al., 2018; Ma & Ying, 2021; Dherin et al., 2022).

We conclude by outlining five domains where our theoretical framework may offer significant implications beyond a mechanistic explanation of in-context learning.

First, our results (Theorems 2.2 and B.2) provide a unifying theory connecting ICL to model editing. We demonstrate that *steering vectors* (Ilharco et al., 2022; Hendel et al., 2023; Todd et al., 2023) and *rank-1 factual model edits* (Meng et al., 2022; Mitchell et al., 2022) naturally emerge as distinct aspects of a single phenomenon: the implicit low-rank weight modulation induced by context. This suggests that these heuristic editing techniques are actually intervening on the same fundamental mechanism that the model uses for self-adaptation.

Second, factorization formulas such as Eq. 31 in Appendix E, derived from Theorem 2.2, rigorously map prompt segments to linear operators and textual concatenation to operator composition. We believe that inspecting the algebraic properties of these operators—such as invertibility and commutativity—lays the groundwork for a formal theory of prompt engineering, moving beyond trial-and-error to a principled understanding of prompt interaction.

Third, because our theory enables the extraction of the exact meta-gradient associated with generation (Proposition 3.1), it provides a novel tool for mechanistic interpretability. Monitoring these gradients dynamically could yield valuable insights into generation health, potentially serving as an early detection signal for hallucinations or mode collapse.

Fourth, analyzing ICL dynamics across different architectures, as initiated in Appendix D, suggests a new direction for model design. By evaluating how different "conceptual layers" facilitate or hinder implicit weight updates, our framework can guide the development of more efficient architectures optimized specifically for in-context adaptability.

Finally, our results highlight a fundamental distinction between the dynamic nature of attention-based inference and the static nature of standard fine-tuning. As noted in our comparison with Chen et al. (2024), our derived weight updates $\Delta_x W$ depend on the specific query token $x$, suggesting that for general transformer blocks, the context cannot be *exactly* represented by a single, fixed weight update. However, this dependency suggests a promising avenue for practical application: investigating whether these token-dependent updates can be aggregated—for instance, via averaging strategies—to produce a single, static weight update that *approximates* the context for any input. Developing such approximations would bridge the gap between our exact mechanistic description of inference and practical techniques for prompt compression and efficient context reuse.

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

# A  RELATED WORK

**In-Context Learning.**  Large language models have the capability to adapt their output based on information or examples provided in the prompt. Because this occurs during inference, there are no explicit gradient updates or modifications of the model parameters. This emergent capability is called in-context learning (ICL) and it has already been shown to exist for GPT-3 in Brown et al. (2020) for a wide range of NLP tasks. Many works have investigated the behavior of ICL to better understand its underlying mechanisms, often through the lens of meta-learning, or learning-to-learn (Schmidhuber, 1987; Hochreiter et al., 2001; Kirsch & Schmidhuber, 2021). A central question within this research area revolves around the precise nature of the "learning" that takes place during ICL and whether ICL represents genuine few-shot learning or instead serves as a mechanism for task-specific inference steering. For instance, the authors of Reynolds & McDonell (2021) question whether true learning occurs at inference time in ICL, contending that the in-context examples instead help the model retrieve capabilities which were already learned during pretraining. This suggests that no new learning actually takes place at inference time. Specifically, Xie et al. (2022) argues that the examples in the prompt serve only as a form of Bayesian conditioning rather than true learning, and they formalize ICL as Bayesian inference. Supporting this direction, Min et al. (2022) shows that replacing example labels with random labels does not dramatically decrease ICL performance, which bolsters the argument that pretrained capabilities are retrieved from the prompt. Though, revisiting these ideas, Wei et al. (2024) show that these results may in fact vary depending on model size and that larger models do start to actually learn from switched labels within the prompt. Still others (e.g., Raventos et al. (2023)), claim that the emergence of true ICL in large language models seems to be dependent on data diversity during pretraining.

Our approach here focuses less on the ontological nature of ICL and more on the actual computational mechanisms taking place within a transformer as the context is processed. This perspective aligns our work most closely with research that frames ICL as meta-optimization, via implicit weight updates, which we now detail.

**Gradient Descent and Meta-Optimization.**  A prominent hypothesis is that ICL performs a type of meta-optimization or implicit gradient descent, essentially finetuning the model through the forward pass (Garg et al., 2022; Akyürek et al., 2023; von Oswald et al., 2023; Dai et al., 2023; Ahn et al., 2023; Zhang et al., 2024). Building on this, our work investigates how ICL is implemented through implicit weight updates that correspond to the underlying learning dynamics of the transformer. Many theoretical analyses of these learning dynamics rely on simplifying assumptions such as single-head, linear attention transformers and prompts formatted as input-output examples determined by a fixed function class; e.g., linear regression (von Oswald et al., 2023; Ahn et al., 2023; Zhang et al., 2024). In particular, von Oswald et al. (2023) shows that linear transformers trained on ICL tasks learn to perform updates analogous to gradient descent. Both Zhang et al. (2024) and Ahn et al. (2023) show that transformers can acquire gradient-based algorithms through ICL and prove global convergence to the optimum for tasks like linear regression within this simplified framework. While the formal analysis of Dai et al. (2023) is also derived for linear attention, their empirical results instead focus on large GPT transformers trained on structured language tasks. However, their core conclusion that ICL operates as implicit finetuning resonates strongly with the viewpoint we take on here. The authors claim that ICL can be understood as implicit finetuning, a perspective similar to the one we take in this work. Other works investigating the link between ICL and gradient descent for standard transformers have largely focused on prompts structured as input-output pair examples. For instance, Garg et al. (2022) demonstrates that standard transformers can in-context learn diverse function classes from such examples, achieving performance comparable to least squares, while Akyürek et al. (2023) shows they can emulate explicit learning algorithms like gradient descent and ridge regression. Separately, attempts to develop a theory for ICL in standard transformers without restricting prompt structure, such as the work by Liu et al. (2025), have thus far needed to incorporate other architectural simplifications or analytical approximations to make the analysis tractable.

In contrast to prior work which has often focused on linear attention or specific prompt structures, our theoretical framework is developed within a more general setting. First, our analysis applies directly to standard multi-head self-attention mechanisms, without requiring any linearity assumptions or other architectural simplifications. Second, our theory holds for arbitrary contextual inputs, not being restricted to prompts formatted as input-output examples. We circumvent both these restrictions by

deriving the exact implicit weight update induced by any context, which then allows for a precise characterization of the implicit learning dynamics of ICL.

**Task Vectors and Model Editing.** The concept of a task vector in machine learning was first introduced in Ilharco et al. (2022) to describe a direction in a model's weight space that encodes task-specific information. There task vectors are derived from the difference between pretrained and finetuned model weights and the authors show how these vectors could be arithmetically manipulated to effectively steer a model's output. The term has since been expanded to include vectors applied to a model's activations as well and several studies have sought to quantify the effect of ICL by analyzing its influence on both weight task vectors and activation task vectors; notably Mitchell et al. (2022); Meng et al. (2022); Hendel et al. (2023). Similarly, Todd et al. (2023) identify "function vectors" (FVs) in transformer hidden states, which are compact representations of in-context learned tasks, extracted via causal mediation analysis over specific attention head outputs. These FVs are shown to be causally effective in triggering task execution even in novel contexts, and distinct from simple semantic offsets, suggesting they act as higher-level function references within the model. Theoretically our work aligns with and extends these ideas. Namely, we demonstrate that the effect of the context can be precisely mapped to an update of the transformer's parameters. Specifically, we show that this effect can be realized as a direct modification of the feedforward weights. In architectures with residual connections, this also includes an additive bias modification (Theorem B.2) in the final layer, which is functionally equivalent to adding an activation task vector, connecting our weight-centric view with activation-based model editing perspectives.

Among these, the work of Hendel et al. (2023) is particularly relevant and closely related to our own. They show that a transformer maps in-context examples to an "activation task vector" that encodes the underlying rule of the examples provided in the prompt. Similar to our main result, they find that manually adding this task vector to the model's hidden states during inference on a new input (without demonstrations) produces outputs similar to those obtained by manually modifying activations with that task vector. While their work offers a mechanistic view of ICL, our approach differs by theoretically deriving the specific weight and bias modifications equivalent to processing a prompt in-context. We prove that a transformer modified by this weight adjustment yields outputs on new inputs that are identical to the original model's outputs when provided with the in-context demonstrations. Finally, our results offer a theoretical counterpoint to recent efforts in "context compression," specifically the work of Chen et al. (2024). They investigate whether In-Context Learning (ICL) can be explicitly converted into model weights for linear attention transformers. Crucially, they prove that for standard architectures, such exact conversion is mathematically impossible. To circumvent this, they propose a modified architecture that adds special bias terms to the attention layers, allowing the context to be compressed into these new parameters. Our Theorem 2.2 demonstrates that while attention weights may resist such exact compression in standard architectures, the MLP layers do not. We show that the effect of the context can be mapped exactly to a low-rank update of the MLP weights in standard transformers, suggesting that the feedforward network acts as a natural reservoir for context-dependent weight adaptation.

The low-rank nature of our ICL-induced weight update appears in other works which explore techniques for explicit model editing. Most notably, ROME (rank-1 Model Editing) in Meng et al. (2022) injects factual associations into transformers by applying targeted rank-1 updates to feedforward network weight matrices. Similarly, MEND in Mitchell et al. (2022) learns optimized low-rank decompositions for model edits. While these methods engineer or learn low-rank modifications for explicit model editing, our theoretical results show that these rank-1 updates to feedforward weights naturally arise as the mechanism by which transformers implement in-context learning. See also other related works which explore how model editing (either through modification of weights or model activations) can be used to achieve the results of finetuning without any gradient-based learning (Subramani et al., 2022; Panickssery et al., 2023; Li et al., 2023; Zou et al., 2023; Liu et al., 2024; Todd et al., 2024; Uppaal et al., 2024; Yang et al., 2025).

# B   CONTEXTUAL BLOCKS WITH SKIP-CONNECTIONS

We now consider the case of contextual blocks with skip connections encompassing the standard Pre-LN transformer block as for instance described in He & Hofmann (2024).

**Definition B.1.** *A contextual block $\Phi$ with skip connection is a layer of the form*

$$\Phi_{W,b'}(C,x) = A(C,x) + W'g_\theta(WA(C,x) + b) + b' \tag{12}$$

*where $g_\theta$ is any differential model parameterized by $\theta$ and $A(C,x)$ is a contextual layer.*

Again, here our motivation and prototypical example is taken from the standard transformer architecture where the contextual layer $A(C,x)$ is a multi-head attention block with a skip connection; i.e.,

$$A(C,x) = x + \text{MultiHeadAttn}(C,x).$$

We can generalize Theorem 2.2 to this context by allowing to update not only the weight matrix $W$ of the first layer but also the bias term $b'$ of the last layer.

**Theorem B.2.** *Consider a contextual block $\Phi$ with skip connection as above; i.e.,*

$$\Phi_{W,b'}(C,x) = A(C,x) + W'g_\theta(WA(C,x) + b) + b' \tag{13}$$

*where $A(C,x)$ is a contextual layer and $g_\theta(z)$ is a differentiable model. Then the effect of a portion $Y \subset C$ of the context $C$ on the output of $\Phi$ implicitly corresponds to a rank-1 weight update of the first-layer weight matrix $W$ given by $\Delta_x W(Y)$ as well as an update of last-layer bias $b'$ given by $\Delta_x b'(Y)$. That is,* *when $A(C\backslash Y, x) \neq 0$, we have that*

$$\Phi_{W,b'}(C,x) = \Phi_{W + \Delta_x W(Y), b' + \Delta_x b'(Y)}(C\backslash Y, x), \tag{14}$$

*and these updates are defined by the following formulas*

$$\Delta_x b'(Y) \quad := \quad \delta A_x(Y), \tag{15}$$

$$\Delta_x W(Y) \quad := \quad \frac{(W\delta A_x(Y))A(C\backslash Y, x)^T}{\|A(C\backslash Y, x)\|^2}, \tag{16}$$

*where $\delta A_x(Y) := A(C,x) - A(C\backslash Y, x)$ is the context vector associated to $Y$. Note that $\Delta_x W(Y)$ is rank-1, since $W\delta A_x(Y)$ is a column vector and $A(C\backslash Y, x)^T$ is a row vector.*

*Proof.* Again, the result follows by direct computation. In the notation above, we have by definition that

$$
\begin{aligned}
\Phi_{W + \Delta_x W(Y), b' + \Delta_x b'(Y)}(C\backslash Y, x) \quad = \quad & A(C\backslash Y, x) \\
& + W'g_\theta\left((W + \Delta_x W(Y))A(C\backslash Y, x) + b)\right) \\
& + b' + \Delta b'(Y) \\
= \quad & A(C\backslash Y, x) + \Delta_x b'(Y) \\
& + W'g_\theta\left(WA(C\backslash Y, x) + \Delta_x W(Y)A(C\backslash Y, x) + b)\right) \\
& + b'
\end{aligned}
$$

Now replacing $\Delta_x W(Y)$ by its definition in Eq. 16 and using that $\frac{z^T}{\|z\|^2}z = 1$, we have that

$$\Delta_x W(Y)A(C\backslash Y, x) = \frac{(W\delta A_x(Y))A(C\backslash Y, x)^T}{\|A(C\backslash Y, x)\|^2}A(C\backslash Y, x) = W\delta A_x(Y).$$

Therefore, simplifying the above and substituting Eq. 15, we get that

$$
\begin{aligned}
\Phi_{W + \Delta_x W(Y), b' + \Delta_x b'(Y)}(C\backslash Y, x) \quad = \quad & A(C\backslash Y, x) + \delta A_x(Y) \\
& + W'g_\theta\left(W(A(C\backslash Y, x) + \delta A_x(Y)) + b)\right) + b'.
\end{aligned}
$$

Since by definition of the context vector we have that $A(C\backslash Y, x) + \delta A_x(Y) = A(C,x)$, we finally get that

$$
\begin{aligned}
\Phi_{W + \Delta_x W(Y), b' + \Delta_x b'(Y)}(C\backslash Y, x) \quad = \quad & A(C,x) + W'g_\theta\left(WA(C,x) + b)\right) + b' \\
= \quad & \Phi_{W,b'}(C,x)
\end{aligned}
$$

which ends the proof. $\qquad\square$

Observe that the bias vector update $\Delta_x b'(Y)$ bears some similarity in spirit with the function vectors of Todd et al. (2024), the transcoder outputs of Ameisen et al. (2025), or the latent concept representations of Hong et al. (2025) used to edit transformer weights. Note also that this theorem is not only valid for contextual layers like Pre-LN transformer blocks as in He & Hofmann (2024) but also other types of contextual layers as, for instance, those in the Griffin recurrent models with local attention De et al. (2024).

## C  STACKING TRANSFORMER BLOCKS

We now explain how to compute the implicit weight updates in the case of a stack of transformer blocks with residual connection as in Theorem B.2. Let us introduce some notation. Let denote by $\Phi^{(i)}$ the $i^{th}$ transformer block. Consider a prompt $C$ a sub-prompt $Y \subset C$ and a token $x^{(0)} \in C\backslash Y$ part of the prompt $C$ but not part of the subset $Y$ that we wish to transfer into the weights. We will denote by $x^{(i)}$ the activation corresponding to the token $x^{(0)}$ after layer $\Phi^{(i)}$, that is

$$x^{(i)} = \Phi^{(i)}(C, x^{(i-1)}). \tag{17}$$

We now explain that we can remove the sub-context $Y$ and patch the MLP weights with our implicit updates by an iterative application of Theorem B.2 from the first layer to the last. Let us start with the first layer. Theorem B.2 tells is that

$$x^{(1)} = \Phi^{(1)}_{W^{(1)}, b^{(1)}}(C, x^{(0)}) = \Phi^{(1)}_{W^{(1)}+\Delta W^{(1)}, b^{(1)}+\Delta b^{(1)}}(C\backslash Y, x^{(0)}), \tag{18}$$

where $W^{(i)}$ stands for the first MLP weight matrix and $b^{(i)}$ for the last bias of the $i^{th}$ transformer block. Applying now Theorem B.2 again to the second block we immediately obtain

$$
\begin{aligned}
x^{(2)} &= \Phi^{(2)}_{W^{(2)}, b^{(2)}}(C, x^{(1)}) \tag{19}\\
&= \Phi^{(2)}_{W^{(2)}+\Delta W^{(2)}, b^{(2)}+\Delta b^{(2)}}(C\backslash Y, x^{(1)}) \tag{20}\\
&= \Phi^{(2)}_{W^{(2)}+\Delta W^{(2)}, b^{(2)}+\Delta b^{(2)}}\left(C\backslash Y, \Phi^{(1)}_{W^{(1)}+\Delta W^{(1)}, b^{(1)}+\Delta b^{(1)}}(C\backslash Y, x^{(0)})\right) \tag{21}
\end{aligned}
$$

Continuing this process up to the last layer, we see that we can remove part $Y$ of the context provided that we patch the original weights iteratively from the first layer to the last one as prescribed by Theorem B.2.

Figure 5 verifies numerically the equivalence between a transformer stack output with the full context and the transformer stack output with the partial context but with weight patched.

### C.1  EXPERIMENTS

In this section, we verify our implicit weight updates for a stack of transformer blocks with residual connections and LayerNorm. We employ the same architecture as described in Vaswani et al. (2017); i.e., the $i$-th transformer block $\Phi^{(i)}$ is defined as

$$\Phi^{(i)}(x) = \mathrm{LayerNorm}(y + \mathrm{MLP}^{(i)}(y)), \quad \text{where } y = \mathrm{LayerNorm}(x + \mathrm{MultiHeadAttn}^{(i)}(x)).$$

Thus, the entire network of $L$ layers can be expressed as

$$\Phi = \Phi^{(L)} \circ \cdots \circ \Phi^{(1)}.$$

Here, we have suppressed the notational dependence on the weight $W^{(i)}$ and bias $b^{(i)}$ for each block $\Phi^{(i)}$ but recall, for each $\mathrm{MLP}^{(i)}$, we have

$$\mathrm{MLP}^{(i)}(x) = W_2^{(i)}\mathrm{ReLU}\left(W_1^{(i)}x + b_1^{(i)}\right) + b_2^{(i)}.$$

If we let $\mathbf{W} := \{W_1^{(1)}, \ldots, W_1^{(L)}\}$ and $\mathbf{b} := \{b_2^{(1)}, \ldots, b_2^{(L)}\}$; then, following the method described above, we can compute the necessary implicit weight updates per layer so that, for a prompt $C$, a sub-prompt $Y \subset C$, and a token $x^{(0)} \in C \setminus Y$, we have

$$\Phi_{\mathbf{W},\mathbf{b}}(C, x^{(0)}) = \Phi_{\mathbf{W}+\mathbf{\Delta W},\mathbf{b}+\mathbf{\Delta b}}(C \setminus Y, x^{(0)}). \tag{22}$$

To verify Equation 22, we run experiments similar to those in Section 4 using the same linear regression datasets, modifying only the transformer architecture to now include the residual skip connection and LayerNorm for both the multi-head attention block and the MLP block, and for a stack of such transformer blocks; see Vaswani et al. (2017).

For our experiments, we take $L = 10$, $d = 2$, $N = 100$, $d_{\mathrm{mlp}} = 128$, $d_{\mathrm{model}} = 32$ and $d_k = d_{\mathrm{model}}/h$ with number of heads $h = 8$. Figure 5 (left, middle) compares the validation loss obtained by using

each side of Equation 22 above to perform evaluation; note here we take $Y = C$. The loss values for both are reported for each checkpoint obtained during training. We can see that these losses are the same for the two computations. Furthermore, we also compare how the intermediate block outputs compare (measured via the $L^2$-norm of the difference) when computed using the original model on $(C, x)$ vs the modified model evaluated using only $x$. That is, we compute

$$\|\Phi_{\mathbf{W},\mathbf{b}}^{(i)}(C, x) - \Phi_{\mathbf{W}+\boldsymbol{\Delta}\mathbf{W},\mathbf{b}+\boldsymbol{\Delta}\mathbf{b}}^{(i)}(x)\|_2, \text{ for } i = 1, \ldots, L.$$

Note that these intermediate block outputs agree with a very high degree of precision, up to order $10^{-6}$, for each intermediate layer of the multi-layer transformer; see Figure 5 (right).

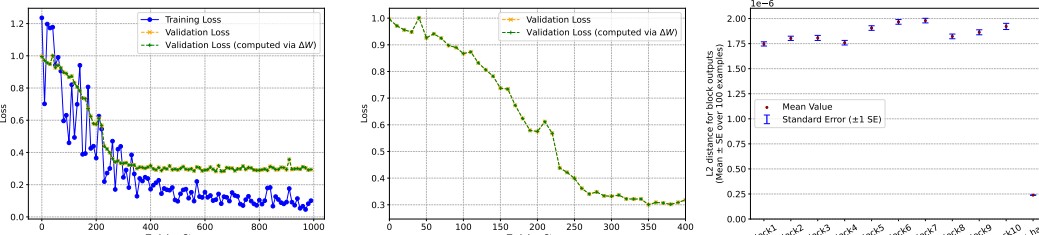

Figure 5: Train and Validation loss curves for multi-layer transformer with LayerNorm. Here, the "Validation loss (computed via $\Delta W$)" refers the loss computed using $\Phi_{W+\Delta W}$; i.e., the trained model prediction given only $x_{\text{query}}$ but with MLP weights modified by $\Delta W$ as defined in Eq. 2. **Left:** Training loss and both validation Loss curves. **Middle:** Close-up of validation loss computed both ways; i.e., using $\Phi_W(C, x)$ vs. $\Phi_{W+\Delta_x W}(x)$. **Right:** Once trained, we sample 100 test tasks $(C, x)$ and for each we perform a forward pass computing both $\Phi_{\mathbf{W},\mathbf{b}}(C, x)$ and $\Phi_{\mathbf{W}+\boldsymbol{\Delta}\mathbf{W},\mathbf{b}+\boldsymbol{\Delta}\mathbf{b}}(x)$. We report the mean and standard error of the L2-norm of the difference of the block outputs for each block in the multi-layer transformer. The block outputs agree with a high degree of precision, up to order $10^{-6}$.

## D  COMPARING CONCEPTUAL LAYERS

In this section, we study the implicit learning dynamics from Section 3 when the conceptual layer is not a self-attention layer, but instead a Recurrent Neural Network (RNN). As Figure 6 shows, the dynamics is less stable compared to a self-attention layer contextual layer as in Figure 3 and fails to converge.

### D.1  EXPERIMENTS

In this section, we further verify the weight transfer formulas we derive in Theorem 2.2 using an recurrent neural network (RNN) as the contextual layer. That is, we take $\Phi = \varphi_W \circ \text{RNN}$ where we've replaced the multi-head self attention block with an RNN and $\varphi_W$, just as before, denotes a dense, fully connected two-layer MLP:

$$\varphi_W(x) = W'\text{ReLU}(Wx + b) + b',$$

with $b, b' \in \mathbb{R}^{d \times \text{mlp}}$ and $W \in \mathbb{R}^{d_{\text{mlp}} \times (d+1)}$. We set the feature dimension of the RNN to be $d_{\text{model}} = 64$ and take $d_{\text{mlp}} = 128$. We train $\Phi$ using the same linear regression dataset tasks as before, with $N = 200$ and $d = 2$. During training we use a batch size of 32 and a learning rate of 0.005 and train for $10,000$ steps. Every 100 steps we report the validation loss on a hold out set for the original model model given the full context $\Phi_W(C, x)$, as well as for the modified model without the context $\Phi_{W+\Delta_x(W)}(x)$. The left and middle images in Figure 6 show the training and validation loss throughout training. Similar to what we saw in Figure 2 when using attention as the contextual layer, the weight transfer is effective in replacing the context $C$ with an exact weight update so that

$$\Phi_W(C, x) = \Phi_{W+\Delta_x W(C)}(x).$$

However, interestingly, we see that the nature of the weight updates as elements of the context are processed is inherently different when using an RNN for the contextual layer as compared to the similar plot produced by using multi-head self attention for the contextual layer; compare Figure 3 and the plot on the right in Figure 6.

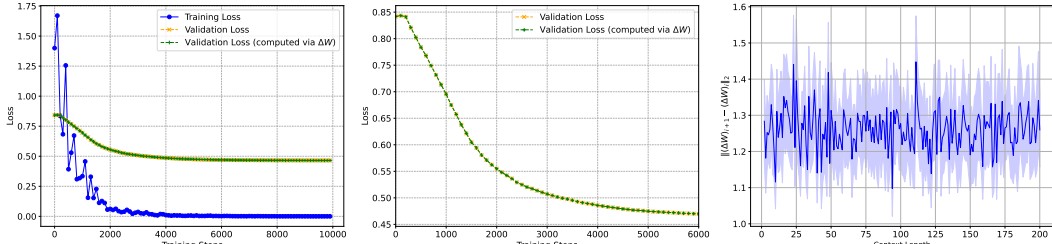

Figure 6: Train and Validation loss curves, $\Delta W$ convergence for an RNN contextual layer. Here, the "Validation loss (computed via $\Delta W$)" refers the loss computed using $\Phi_{W+\Delta W}$; i.e., the trained model prediction given only $x_{\text{query}}$ but with MLP weights modified by $\Delta W$ as defined in Eq. 2. **Left:** Training loss and both validation Loss curves. **Middle:** Close-up of validation loss computed both ways; i.e., using $\Phi_W(C,x)$ vs. $\Phi_{W+\Delta_x W}(x)$. **Right:** As more of the context in processed, the relative change in the weights $W$ fails to converge when using an RNN as a contextual layer. For context length $i > 2$, the plots shows the average difference $\|(\Delta W)_{i+1} - (\Delta W)_i\|_2$ and the standard error over 100 separate trials.

## E  AN ALTERNATIVE IMPLICIT LEARNING DYNAMICS OF ICL

In this section, we describe an alternate view on the implicit learning dynamics which follow from an iterative application of Theorem 2.2.

This approach differs in that it interprets how each context token input of a transformer affects the contextual block output. It's based on the idea that the influence of each context token on the model's output can be seen as an implicit change in its behavior. While the transformer's weights are not actually updated as it generates a response, the final output is effectively the same as if the model had undergone a rapid learning process influenced by the context. We will now describe this implicit learning dynamic.

This approach differs in that it uncovers the implicit dynamics generated by the effect of each context token on the contextual block output. As a result, this means that while no explicit weight update is performed while a transformer block generates the first response token, the actual output is equivalent to that of the contextual block without context but for which an implicit learning dynamics in weight space has happened. We now describe in detail these learning dynamic.

Starting with the initial weight $W_0$ for the first dense layer of the neural network, we have

$$\Phi_{W_0}(c_1,\ldots,c_n,x) = \Phi_{W_0+\Delta W_0(c_1)}(c_2,\ldots,c_n,x) \tag{23}$$

which gives us the first weight update corresponding on the effect of token $c_1$ on the first-layer weight matrix:

$$W_1 = W_0 + \frac{(W_0\Delta A(c_1))A(c_2,\ldots,c_n,x)^T}{\|A(c_2,\ldots,c_n,x)\|^2} \tag{24}$$

If we continue this process iteratively, we obtain the next weight update corresponding to the consumption of the second token:

$$T_{W_1}(c_2,\ldots,c_n,x) = T_{W_1+\Delta W_1(c_2)}(c_3,\ldots,c_n,x) \tag{25}$$

which yields

$$W_2 = W_1 + \frac{(W_1\Delta A(c_2))A(c_3,\ldots,c_n,x)^T}{\|A(c_3,\ldots,c_n,x)\|^2} \tag{26}$$

We can summarize this iterative process of implicit weight updates for each successive token:

**Corollary E.0.1.** *In the notation above, the iterative process of weight updates*

$$W_i = W_{i-1} + \frac{(W_{i-1}\Delta A(c_i))A(c_{i+1},\ldots,c_n,x)^T}{\|A(c_{i+1},\ldots,c_n,x)\|^2} \tag{27}$$

*starting with the initial weights of the first dense layer $W_0$ models the transfer of information from the prompt token $c_i$ into the contextual block weights: Namely, we have that*

$$T_{W_i}(c_{i+1},\ldots,c_n,x) = T_{W_0}(c_1,\ldots,c_n,x), \tag{28}$$

*for all $i = 1, \ldots, n$ with $\Delta A(c_i) = A(c_i, \ldots, c_n, x) - A(c_{i+1}, \ldots, c_n, x)$.*

Notice that $\Delta A(c_i)$ measures the effect of context token $c_i$ on the contextual block output. When $c_i$ has no effect on the output, that is when $\Delta A(c_i)$ is zero, and the corresponding update vanishes. Notice that the weight update at step $i$ is linear in the weights; namely, we can rewrite it as

$$W_i = W_{i-1} + h_i W_{i-1} A_i = W_{i-1}(1 + h_i A_i) \quad \text{where} \quad A_i := \Delta A(c_i) A(c_{i+1}, \ldots, c_n, x)^T \quad (29)$$

with adaptive learning rate given by

$$h_i := \frac{1}{\|A(c_{i+1}, \ldots, c_n, x)\|^2}. \tag{30}$$

In particular, this gives us a factorization formula for the total implicit weight matrix corresponding to the effect of context $[c_1, \ldots, c_n]$ on input-token $x$:

$$W_n = W_0(1 + h_1 A_1)(1 + h_2 A_2) \cdots (1 + h_n A_n). \tag{31}$$

# F  LLM USAGE DISCLOSURE

We used Gemini 2.5 Pro to assist in polishing the grammar and improving the clarity of the final draft. The authors reviewed and edited all generated text to ensure correctness.

