# OpenReview forum: "Learning without training: The implicit dynamics of in-context learning"
_ICLR.cc/2026/Conference — Submitted to ICLR 2026_

### Official Review · Reviewer_bp19 · 2025-10-31

**Soundness:** 4
**Presentation:** 4
**Contribution:** 4
**Rating:** 8
**Confidence:** 3

**Summary:**

This work proves that the impact of context tokens on the output of a contextual block (a generalization of a transformer block) is equivalent to a rank-1 update of the weight matrix of the feedforward component. This result explicitly links in-context learning to a form of gradient descent. Additionally, this work proves that in-context learning converges as context length increases. Both of these theoretical results are demonstrated experimentally for a 1 layer transformer.

**Strengths:**

The theoretical results are clear, compelling, and are of broad interest to the community.

The experimental results perfectly correspond to the theory.

The equivalent rank-1 weight update enables explicit comparisons between in-context learning and finetuning, which can enable future work.

**Weaknesses:**

Extending the experimental results to other forms of contextual layers would be of interest, especially as the authors note that different types of contextual layers might be more or less effective in-context learners. Though potentially out of scope, a comparison of the weight updates between e.g., RNN and attention components would be of broad interest.

Furthermore, any intuitions regarding why finetuning is more efficient in the 5-20 step regime would strengthen the work and help to flesh out that section.

**Questions:**

See weaknesses section.

---

> ### Author Response · Authors · 2025-11-20
>
> Thanks for your review. We are delighted that you found our soundness, presentation, and contribution excellent, and that our "theoretical results are clear, compelling, and are of broad interest to the community".
>
> You’ll find below our best effort to answer your questions. Please consider increasing your score if you are satisfied with them.
>
> > *Extending the experimental results to other forms of contextual layers would be of interest, especially as the authors note that different types of contextual layers might be more or less effective in-context learners. Though potentially out of scope, a comparison of the weight updates between e.g., RNN and attention components would be of broad interest.*
>
> This is a great point! We now added a replication of our ICL experiment with a conceptual layer that is an RNN (see the new **Appendix D, Fig 6**). Very interestingly, we see that the implicit weight update dynamics are much more unstable when comparing the convergence of the weight updates of the different contextual layers (attention vs RNN). The RNN layers appear to have a much more difficult implicit convergence (see Fig 6 (right)) than those we witness with attention based layers, even though both architectures achieve good test error on the dataset. This discrepancy is interesting but admittedly not well understood! One conjecture is that this is perhaps is due to the fact that classical RNNs (LSTMs/GRUs) struggle with ICL because of a "hidden state bottleneck" while transformers are able to get around this via induction heads and how information is retrieved via attention.
>
> We do agree that this line of experimentation may help understanding the advantage of particular contextual layers over others; however we leave this discussion and further exploration into this line of inquiry for future work.
>
> > *Furthermore, any intuitions regarding why finetuning is more efficient in the 5-20 step regime would strengthen the work and help to flesh out that section.*
>
> This is an interesting observation. While we don’t have a concrete answer to this question, we believe that the behavior of fine-tuning (v.s. ICL) is strongly dependent on the task and the modality of the fine-tuning (i.e. full fine-tuning v.s. efficient fine-tuning / LoRA).
> Except for our experiments, we are not aware of a fine-tuning strategy, targeting the same weights as our theorem (i.e., only the last MLP bias and first MLP weight matrix). In our limited experiments, the dynamics seems relatively similar, but a more thorough investigation of this particular type of fine-tuning v.s. ICL is required. We think this a very interesting line of research and might further shed light on PEFT methods or the role of low-rank updates during fine-tuning, but this is beyond the scope of the current paper and we leave it for future work.

---

### Official Review · Reviewer_DrGD · 2025-11-01

**Soundness:** 3
**Presentation:** 3
**Contribution:** 3
**Rating:** 8
**Confidence:** 3

**Summary:**

This paper shows that in context learning (ICL) can be reframed as low-rank updates to the MLP weights in a transformer. TThey prove that the efect of any subset of the context Y on the block’s output for a query x can be represented exactly by a rank‑1 update to weight matrix W. A corollary shows how the entire context can be folded into W (Eq. 2), and an extension (Theorem B.2) handles residual connections by adding a bias update equal to the context vector . Building on this identity, the authors derive an implicit learning dynamics across tokens that resembles stochastic gradient descent with a token‑dependent linear loss.
Experiments use a single‑block decoder‑only transformer trained on linear regression ICL tasks. They show that predictions obtained with the prompt are numerically indistinguishable from predictions made without the prompt after applying the rank‑1 weight transfer, that the induced updates converge as more tokens are consumed, and that the implicit updates are highly aligned with explicit finetuning gradients

This work gives an elegant identity that reframes ICL as "learning without training" via rank‑1 weight transfers inside a transformer block. The theoretical statement is clean and broadly framed; the experiments validate the identity in a controlled setting and suggest links to finetuning.

**Strengths:**

* The topic is interesting and relevant to the community. I find the results really striking and simple. The closeness of the match in figure 2 are very compelling
* Theorem 2.2 provides an exact algebraic identity showing that a contextual layer followed by an MLP is equivalent to a prompt‑free forward pass with a rank‑1 edit to the first MLP layer. The formula is simple and easy to implement.
* This generalizes beyond linear attention, which I think sets it apart in related work that I am familiar with
* I appreciate the discussion on the connections between steering, ICL, and ROME style updates that this work brings together nicely. I think this will be useful for the community

**Weaknesses:**

* Experiments are confined to linear regression ICL with a small, single‑block model without LayerNorm or MLP residuals (the latter are treated theoretically in App. B but not empirically). I think it would be of great interest to see the empirical results with the residuals at least.
* While the analysis is pretty deep, I wish the authors would be able to say something more about the updates to stacked blocks and how they interact. Is there even a heuristic approximation the authors can do to compare empirically?
* I think this paper would be much more impactful if it could say something about the generation beyond the first token. Benefits of ICL accrue over rollouts, and I feel like theorem

**Questions:**

Given an MLP (that follows an attention block) with two weight matrices, W1 and W2, and a relu between them, I'm still a little confused whether this is able to say anything about how W2 changes?

---

> ### Author Response · Authors · 2025-11-20
>
> Thank you for your thoughtful review! We are delighted that you found "this work gives an elegant identity that reframes ICL as learning without training". It makes us happy that you think our "theoretical statement is clean and broadly framed"  and that this "will be useful for the community". We agree. We are glad that you find the results "really striking and simple" and that "the closeness of the match in figure 2 are very compelling". We spend a lot of time finding the right formulation for the statement so that the result appears as broad and simple as possible, and we are happy you appreciate this.
>
> You’ll find below our best effort to answer your questions.
>
> Please consider increasing your score if you are satisfied with them.
>
> > *Experiments are confined to linear regression ICL with a small, single‑block model without LayerNorm or MLP residuals (the latter are treated theoretically in App. B but not empirically). I think it would be of great interest to see the empirical results with the residuals at least.*
>
> Good point! We now include experiments with residual, several transformer blocks, and LayerNorm in **Appendix C** and **Fig. 5** in the revised version.
>
> > *While the analysis is pretty deep, I wish the authors would be able to say something more about the updates to stacked blocks and how they interact. Is there even a heuristic approximation the authors can do to compare empirically?*
>
> In fact, the case of stacking transformer blocks follows immediately by a simple induction. We have now added a detailed discussion on this in the **Appendix C and Fig. 5**, and replicated the experiments with several blocks. Thank you for this suggestion!
>
> > *I think this paper would be much more impactful if it could say something about the generation beyond the first token. Benefits of ICL accrue over rollouts, and I feel like theorem*
>
> We agree with you. We added this in **Corollary 2.6.1**. In fact, our theorem can be used to compute weight updates beyond the first generated token. This is only a matter of choosing what part of the context we want to remove appropriately. There are several ways to deal with this, but here is a particularly simple one. Consider the prompt $[Y, x_1, \dots, x_i]$, where $x_1,\dots, x_i$ are the generated tokens and Y is the context provided by the user. Then our theorem gives the update for the $i$ token as $T_{W + \Delta_{x_i} W(Y)}(x_1, \dots, x_i) = T_{W}(Y, x_1, \dots, x_i)$ . Analyzing the implicit update $\Delta_{x_1} W_1(Y),\dots, \Delta_{x_n} W_n(Y)$ may provide useful information as to where the generation is going. We now added a footnote on p.2, a new **Corollary 2.6.1**, and a paragraph on this in the conclusion (**lines 519-522**).
>
> Note: it seems your final thought may have been cut off and not completed. The second sentence: "Benefits of ICL accrue over rollouts, and I feel like theorem…". Is there more you wanted to express here? Also, can you clarify what you mean by "Benefits of ICL accrue over rollouts"? In what sense do the benefits accrue (it also seems the sentence is not complete)?
>
> > *Given an MLP (that follows an attention block) with two weight matrices, W1 and W2, and a relu between them, I'm still a little confused whether this is able to say anything about how W2 changes?*
>
> Good question. Our main Theorem 2.2  shows how the initial MLP layer with weight $W_1$ is able to absorb the effect of the change in context from the preceding contextual layer. This works very well mathematically but also lends itself to a nice interpretation given recent work in understanding the role of MLP blocks in LLMs. Namely, Geva et al. (2021) have identified these MLPs blocks as information storages in the form of soft key-value stores. The neuron vectors of the first matrix $W_1$ correspond to the keys, while the neuron vectors in $W_2$ correspond to the values. Within this interpretation of the MLP blocks, our formula shows that only the keys are implicitly updated during prompt consumption. We now added a discussion about this in the revised version in new **Remark 2.5**.

---

### Official Review · Reviewer_NRGH · 2025-11-01

**Soundness:** 2
**Presentation:** 2
**Contribution:** 1
**Rating:** 2
**Confidence:** 4

**Summary:**

This paper attempts to understand in-context learning by showing that context data can be interpreted as an implicit low-rank update to the MLP weights.
Specifically, the authors introduce the concept of a contextual block and show that adding context tokens is mathematically equivalent to applying a rank-1 modification to the MLP weight matrix. They further show that processing the context sequentially induces a gradient-descent-like implicit learning dynamics.

**Strengths:**

The work provides an interesting angle to understanding ICL, which appears to be less restrictive and more general than existing works, e.g., doesn't rely on attention linearization. The writing is clear and easy to follow. Theoretical analysis is supported by empirical verification.

**Weaknesses:**

### Limited significance
The idea sounds interesting but in practice it is mostly rewriting the MLP function with some basic algebra.  When considering the fact that the MLP does not mix tokens, including the separate context tokens does not seem as meaningful.

Following the same context data, we could have different questions or instructions. Ideally, the parameter updates associated with context data should be independent to question/instruction tokens afterwards. In the paper's notation, \Delta_x W should be independent of x. In a related work [1], this is achieved by considering attention linearization and the corresponding weight update of each context token is also a rank-1 matrix in the attention "bias".
However, if my understanding is correct, this is not the case in Thm 2.2 and the results seem less significant.

The weaknesses of only affecting the first token is one that has been prevalent in the literature.  Before it is solved, this can only provide rough intuition as to how context can be designed rather than concrete improvements.

### Limited scope and practical relevance
The work provides implicit dynamics of ICL within transformers.
Even though the paper claims to be more generally applicable to different LLM architectures with fewer restrictions, it is only valid for a single transformer block hence the limited scope.
On the other hand, the theoretical analysis does not seem to highlight what is the meaning behind such dynamics and how they can be taken advantage of moving forward in some practical scenarios.


[1] Exact Conversion of In-Context Learning to Model Weights in Linearized-Attention Transformers. ICML 2024

**Questions:**

* Given the weight change of the MLP varies once you have more than a single input  (as the input forms part of the context), is there a tangible direction forward which can possibly remedy this issue?

* What tangible benefits might arise from understanding these implicit dynamics beyond the providing an alternative to understanding ICL?

---

> ### Author Response · Authors · 2025-11-20
> **Part 1**
>
> Thank you for your detailed review. We are very happy that you found that our "work provides an interesting angle to understanding ICL, which appears to be less restrictive and more general than existing works". Also, we are glad you found that the "writing is clear and easy to follow" and that the "theoretical analysis is supported by empirical verification".
> You’ll find below our best effort to answer your questions.
>
> Please consider raising your score if satisfied.
>
> > *Following the same context data, we could have different questions or instructions. Ideally, the parameter updates associated with context data should be independent to question/instruction tokens afterwards. In the paper's notation, $\Delta_x W$ should be independent of x. In a related work [1], this is achieved by considering attention linearization and the corresponding weight update of each context token is also a rank-1 matrix in the attention "bias". However, if my understanding is correct, this is not the case in Thm 2.2 and the results seem less significant.*
> [1] Exact Conversion of In-Context Learning to Model Weights in Linearized-Attention Transformers. ICML 2024 (link)
>
> We certainly missed [1]. Thanks for pointing it out to us, since it is very interesting and definitively relevant to the current work, and we now cite it in our introduction (**lines 073-083**) as well as in our Related Work of the revised version (**lines 785-794**) .
>
> Also, you raise an interesting and correct point: The implicit MLP weight updates that *exactly* reproduce the effect of the context C are indeed different for each given input token x.
>
> That said, we want to stress here that the goal of the current paper is different from the goal of [1]. Our goal is *not* to convert a prompt into reusable weight for a practical purpose. Our goal is to understand the mechanics of what happens when a trained transformer consumes context at inference time. What we have found is theoretically interesting, as it tells us that the *exact* effect of the context on the generated token can not be *exactly* represented by a fixed weight update in the same way as would be produced by fine-tuning.
>
> However (while again the goal of this paper is not to compress a prompt into weights) we believe our approach may lead to progress into this important question. Namely, it is possible to *approximate* the effect of the context on the generated token by a single implicit update of the same form, using the token-dependent updates of our theorem in a sort of averaging strategy. While this is very interesting and points towards a promising future research direction, it is out of scope for the present paper. Our current work aims at establishing the foundational theory which explicitly quantifies the effect of the context through in-context learning via implicit weight updates.
>
> We do not believe at this point that implicit weight updates *exactly* replicating the effect of the context can be found for generic transformer blocks (i.e.; without linear attention assumption). However, our results do provide another direction in which it is possible to compute useful approximate ones. We added a paragraph now discussing this in the conclusion (**see lines 527-535**).

---

> ### Author Response · Authors · 2025-11-20
> **Part 2**
>
> > *The idea sounds interesting but in practice it is mostly rewriting the MLP function with some basic algebra.*
>
> Indeed! It may seem shocking that such a general insight boils down to basic linear algebra. However, this simplicity is possible *precisely* because we have framed the essential concepts at the right level of abstraction. Rather than focusing on simplifications or approximations of the attention mechanism itself, we instead abstract this component as a contextual layer and focus on showing (through only some basic algebra) how the weights of the MLP can be made to absorb that difference. This allows for a flexible and general statement (no restriction on the self-attention layer). We feel this is precisely the value of our approach: the simplicity of the proof is not a weakness and is, in fact, a strength and a consequence of our careful formulation.
>
> To summarize, the insight of our approach and proof are as follows:
> * an understanding of the self-attention layer as producing context vectors $\delta A_x(Y) = A(C,x) - A(C\backslash Y, x)$
> * realizing that the affine maps can absorb the effect of these context vectors into their weights in a precise and exact way, without the need for approximation or the special case of linear attention.
>
> Once this has been understood, our result follows easily and elegantly for even the most general case (without any additional assumptions on the nature of the contextual layer such as linear attention, single attention head, etc)
>
> In the revised manuscript, we demonstrate this extensibility by generalizing our results to **standard multi-layer Transformer blocks** (including multiple layers, Residuals and LayerNorm; see **Appendix C, Fig. 5**) and other contextual layers such as **recurrent architectures** (e.g., RNNs; see **Appendix D, Fig. 6**). In both settings, our formulas are highly precise and lend interesting insight to the nature of how context is processed by these models.
>
> > *When considering the fact that the MLP does not mix tokens, including the separate context tokens does not seem as meaningful.*
>
> We appreciate your feedback and would love to clarify any misunderstandings. Could you please rephrase your concern regarding "including separate context tokens"? We'd like to understand your perspective on what aspects of our approach you find less meaningful and why, so we can properly address them and provide a thorough and helpful response.
>
> > *The weaknesses of only affecting the first token is one that has been prevalent in the literature. Before it is solved, this can only provide rough intuition as to how context can be designed rather than concrete improvements.*
>
> This is a good point! In fact, our theorem is valid for any token that is not within the part of the context we want to remove (i.e., for any $x$  not in $Y$ in the notation of Theorem 2.2) and not only for the first token we want to generate. We have now clarified this in the current revision (see footnote on page 2). In particular, we can apply our theorem to the $n^{th}$ token generated by setting $C = [Y, x_1, \dots, x_n]$ where the $x_i$’s are the generated tokens and $Y$ is the user input, since the theorem is valid for any of $x_1$, $\dots$, $x_n$. See also the new **Corollary 2.6.1** which further clarifies this point.
>
> > *The work provides implicit dynamics of ICL within transformers. Even though the paper claims to be more generally applicable to different LLM architectures with fewer restrictions, it is only valid for a single transformer block hence the limited scope.*
>
> In fact our theorem generalizes directly for a stack of transformer blocks via simple induction by sequentially applying the theorem from the first block to the last, and computing the deltas using the activations with and without context at each step. Please see the **new Appendix C** in the revision which further provides more details and verifies our formulas experimentally (see **Figure 5**) for a stack of $L$ transformer blocks each with residual connection and LayerNorm; i.e., a stack of transformer blocks of the form $\Phi^{(i)}(x) = \text{LayerNorm}(y + \text{MLP}^{(i)}(y)), \quad \text{where } y = \text{LayerNorm}(x + \text{MultiHeadAttn}^{(i)}(x)).$

---

> ### Author Response · Authors · 2025-11-20
> **Part 3**
>
> > *On the other hand, the theoretical analysis does not seem to highlight what is the meaning behind such dynamics and how they can be taken advantage of moving forward in some practical scenarios.*
>
> We believe the theoretical implications of our work are significant, and uncover a dynamic understanding of the implicit "learning" that occurs during inference. Traditionally, learning has been understood as explicit weight updates derived from loss gradients. However, LLMs have demonstrated in-context learning capabilities without these explicit updates. Our findings aim to demystify this phenomenon and we prove that transformer computations are mathematically *equivalent* to implicit loss updates at inference time. We also reveal that these resulting learning dynamics bear a strong resemblance to the explicit learning that occurs via fine-tuning.
>
> The implicit dynamics that we uncover in this work point to many practical implications which take advantage of this insight. As a first example, consider optimization. Just as gradients monitor learning health during training, our derived relationship can be used to inspect the implicit learning dynamics during inference. Other potential directions include applications to model editing, a formal understanding of the role of prompt engineering, and a better evaluation of various conceptual layer architectures.
>
> Thank you for this feedback. We have updated the revised conclusion (see **lines 506 to 535**) to discuss in more detail these potential implications of our work. However, their full exploration remains the goal of future research.
>
> > *Given the weight change of the MLP varies once you have more than a single input (as the input forms part of the context), is there a tangible direction forward which can possibly remedy this issue?*
>
> This is a very compelling question and great point. As you correctly note, within our current theory it is not possible to create a single global weight update which is *exact* since the implicit weight update would be different for different inputs. However, we believe this can be achieved by *approximating* this effect through aggregating the implicit updates per token. A rigorous theoretical treatment in this direction involves significant complexity and is beyond the scope of the current work. We  now discuss that in the conclusion (see **lines 506-535**).
>
> > *What tangible benefits might arise from understanding these implicit dynamics beyond the providing an alternative to understanding ICL?*
>
> As mentioned in our previous response, one such tangible benefit lies in translating abstract ICL behaviors into quantifiable metrics. By establishing a *precise* formula for the implicit gradient, this work enables useful inference-time diagnostics. Just as one monitors training gradients for stability during traditional training, one can now inspect the implicit weight evolution to potentially identify the precursors of undesirable behaviors like hallucinations, recency bias, or prompt format sensitivity. Although this is beyond the scope of this paper, we see this as a crucial step for the field of mechanistic interpretability and have added a detailed discussion of these downstream applications to the conclusion (see **lines 506 - 535**). As an example, we see in the new **Appendix D, Fig 6**, that contextual layers based on RNN have a less stable and less convergent implicit learning dynamics compared to attention-based contextual layers.

---

### Official Review · Reviewer_iGAG · 2025-11-04

**Soundness:** 1
**Presentation:** 2
**Contribution:** 1
**Rating:** 2
**Confidence:** 4

**Summary:**

The paper claims that the effect of an in-context prompt on a transformer block can be written as an equivalent rank-1 “implicit weight transfer” to the first MLP layer (and a bias tweak in a residual variant). Theorems 2.2 / B.2 state this equivalence; Proposition 3.1 describes an “online gradient” view. Experiments verify near-exact numerical agreement on a toy linear-regression ICL setup.

**Strengths:**

The paper is easy to follow.

**Weaknesses:**

The main results reduce to short, fairly obvious linear-algebraic identities; proofs are a few lines and mostly restate the construction. Novelty and significance are limited; empirical evidence is restricted to a tiny, non-representative setting; and there are correctness/notation issues that need fixing, e.g.:
1. Unstated non-degeneracy needed for the update formulas. Both the main theorem and its residual variant divide by $\lVert A(C\setminus Y, x)\rVert^2$ and the corollary by $\lVert A(x)\rVert^2$ without assuming these are nonzero. The proofs also apply the identity $\frac{z^\top}{\lVert z\rVert^2} z = 1$, which requires $z\neq 0$.
2. You present the update $\Delta W = \dfrac{(W\delta A_x(Y))A(C\setminus Y,x)^\top}{\lVert A(C\setminus Y,x)\rVert^2}$ as if unique. In fact, any $\Delta W + M \text{ with } MA(C\setminus Y,x)=0$ yields the same effect.

**Questions:**

see weaknesses.

---

> ### Author Response · Authors · 2025-11-20
> **Part 1**
>
> Thank you for your review. We address your specific questions and concerns point by point below and hope this clarifies any confusion with our work.
>
> If you are satisfied with our answers, please consider raising the score.
>
> > *The main results reduce to short, fairly obvious linear-algebraic identities; proofs are a few lines and mostly restate the construction*
>
> We understand that it can feel surprising that such a general statement (no restriction on the self-attention layer) can be derived so directly. However, this post-hoc simplicity is only evident after finding the right level of abstraction, which is the value of our approach and which seems to have been missed here.
>
> The key insight in our proof is not so much the algebraic manipulation (for which we made a concerted effort to make as simple as possible), but rather the identification of the right concepts that allow such a simple computation to prove such a general statement.
>
> As we describe in the related work section, previous attempts to model context as implicit weight modification (e.g., Oswald et al., 2023; Zhang et al., 2024; Ahn et al., 2023, Dai et al., 2023,  Chen et al.(2024)) have focused on *attention* weights and largely relied on approximations—assuming linear attention, single heads, or specific prompts. The real insight of our approach and proof are twofold:
> * an understanding of the self-attention layer as producing context vectors $\delta A_x (Y) = A(C,x) - A(C\backslash Y x)$
> * realizing that the affine maps can *absorb* the effect of these context vectors into their weights in a precise and exact way, without the need for approximation or the special case of linear attention.
>
> Once this is understood, the result that context acts as an implicit weight update for token generation follows naturally for the general case. This is an important realization.
>
> Lastly, we argue that the fact that our "main results reduce to short, fairly obvious linear-algebraic identities" and that the "proofs are a few lines" actually speak to the elegance, brevity and generality our approach which are typically considered *favorable* qualities of a proof. In fact, this generality is precisely what allows our ideas to be easily translated to more complex settings or other contextual layers. In the revised manuscript, we demonstrate this extensibility by generalizing our results to **standard multi-layer Transformer blocks** (including multiple layers, Residuals and LayerNorm; see **Appendix C, Fig. 5**) and other contextual layers such as **recurrent architectures** (e.g., RNNs; see **Appendix D, Fig. 6**).
>
> > *novelty and significance are limited*
>
> Could you please explain in which sense either the novelty or significance are limited in your opinion? (We are failing to see a justification for these points in your review.)
>
> As far as we are aware, this paper is the first to demonstrate without any limiting assumption on either the transformer block (e.g. linear attention, single attention head) or the type of prompt  (e.g. regression prompts) that the context acts *exactly* as an implicit MLP layer weight update. This insight further allows us to uncover implicit learning dynamics which occur at inference time.
>
> Please tell us what you think is not novel: i.e., Can you please provide a reference for other published work which does the same?
>
> In terms of significance, the emergence of learning at inference time without explicit weight update is possibly one of the main mysteries in LLMs. We believe that providing the mathematical underpinning rendering this possible is actually *extremely*  theoretically significant. We believe the theoretical significance goes even beyond in-context learning. Namely, our framework is a step in understanding as to how tasks and knowledge are represented in the space of model weights. Practically, our framework enables new capabilities, such as monitoring and steering of generation dynamics, as well as compressing context into efficient weight updates, among many other applications.
>
> We appreciate that a clear through line to these applications is missing in the current paper and we now make this clearer in the conclusion of the revised version (**see lines 506-535**).

---

> ### Author Response · Authors · 2025-11-20
> **Part 2**
>
> > *empirical evidence is restricted to a tiny, non-representative setting*
>
> We respectively want to note that our experiments mostly serve as a verification of the math, exploring the impact of the numerical rounding on our results. Since the formula is exact (not an approximation), it does not make a lot of sense to verify the same exact relation in many settings. The only possible source of disagreement is numerical rounding, which we also quantify in our experiments.
>
> That said, we have now expanded our analysis to standard multi-layer architectures, including residual branches and LayerNorm (as in Vaswani et al., 2017). As shown in **Appendix C, Fig 5**, our math and theoretical results scale to these general architectures as well.
>
> As for the setting, we chose the standard regression setting from previous established work (Garg et al., 2022; Akyürek et al., 2023; von Oswald et al., 2023) to ensure direct comparability. Also, since our focus is the *mechanism* of ICL rather than its emergence, this setup is optimal; that is, predicting a single scalar isolates next-token dynamics without the confounding factors of auto-regressive language generation. Furthermore, this setting naturally facilitates a precise comparison between our implicit dynamics and explicit fine-tuning.
>
> > *there are correctness/notation issues that need fixing*
>
> Thank you for noting this. We added in our statements the restriction that the attention $A(C\backslash Y, x)$ and $A(x)$ need to be non-zero for the result to hold. We want to note that this is a reasonable assumption in practice: We also observe this in our experiments, which follow Theorem 2.2. And Corollary 2.3.1, very accurately (see Figure 2 and our new **Figure 5** in the case of several layers).
>
> As for the non-uniqueness of the update, this is an interesting aspect which we now discuss in the revised version (see in red **Remark 2.4**). The existence of equivalent updates is a standard consequence of neural network over-parameterization and does not invalidate our main result which shows that context can be viewed as an implicit weight update on the MLP. However, we emphasize that our specific formulation produces *rank-1 updates*, which are minimal within the space of functionally equivalent solutions.

---

### Author Response · Authors · 2025-11-20
**Thank you to all Reviewers**

**To All Reviewers**

We thank you all for your constructive feedback and for your time and effort reviewing our manuscript.

We are encouraged that you found our work "compelling" and offering "an interesting angle to understand ICL" with results "clear, compelling" and "of broad interest to the community". We are glad that you found them "easy to follow" and that we sufficiently "support our theoretical analysis with empirical verification", which "perfectly corresponds to the theory".

We deeply appreciate your insights and have revised the paper to address your concerns. We believe these changes substantially improve the work.

**Key Revisions:**

* **Appendix C**: Now includes a formal analysis of the multi-layer transformer, including experiments verifying our results
* **Figure 5**: Validates our main findings for transformers with Residual connections and LayerNorm.
* **Appendix D**: Compares RNN vs. Attention contextual layers, highlighting the superior stability of attention-based implicit learning dynamics.
* **Corollary 2.6.1**: Demonstrates that our results extend beyond the first generated token.
* **New Conclusion:** We now detail tangible directions better outlining the significance of our work


We hope that these revisions and our responses below resolve your concerns.

If so, we would appreciate it if you would consider raising your scores.

The Authors

---

### Meta-Review · Area_Chair_8n9Z · 2026-01-06

**Summary:**

The paper proposes a theoretical framework interpreting In-Context Learning (ICL) as implicit low-rank weight updates to the MLP layers of a Transformer block. The authors derive an algebraic identity showing that a forward pass with context is equivalent to a forward pass without context using modified weights. Based on this, they claim the process follows an implicit gradient descent dynamic. The submission includes experiments on linear regression tasks to verify these identities and analyze convergence.

**Reviewer Concerns:**

Addressed Concerns:
- Architecture Generalization: In response to Reviewer DrGD’s request to see results beyond a single block without residuals 4, the authors added Appendix C and Figure 5 to demonstrate the framework on multi-layer architectures with residual connections and LayerNorm.
- Comparative Analysis: Addressing Reviewer bp19’s suggestion , the authors included Appendix D to compare the implicit dynamics of Attention against RNNs.

Outstanding Concerns:
- Theoretical Validity (The "Loss" Disconnect): While the paper proves the process is equivalent to SGD on some formal loss function, it fails to establish any connection between this implicit loss and the actual optimization objective (e.g., L2 loss) used to update MLPs. As noted by Reviewer iGAG, the results reduce to "obvious linear-algebraic identities", and without a link to the physical training objective, the analogy to "learning" is mathematically superficial.
- Input-Dependent Weights: Reviewer NRGH correctly pointed out that the derived weight update $\Delta_x W$ depends on the query token $x$. This contradicts the standard definition of learning or fine-tuning, where weights are fixed parameters independent of the test input. The rebuttal failed to resolve this fatal flaw in the "fine-tuning" analogy.
- Experimental Triviality: Experiment 1 (Figure 2), which shows a match between the ICL output and the implicit update output, is tautological. Since the theoretical derivation is an exact algebraic identity, perfect numerical agreement is expected by definition. This experiment merely validates the correctness of the authors' code, not the scientific hypothesis that "learning" is occurring.

**Reviewer Scores:**

- Reviewer iGAG (Current: 2): Would likely maintain their score. Their critique that the work is essentially "rewriting the MLP function with some basic algebra"  is accurate and reinforced by the lack of a meaningful physical interpretation of the loss.
- Reviewer NRGH (Current: 2): Would likely maintain their score. Their concern about the input dependence of the weight updates remains a fundamental barrier to accepting the "implicit fine-tuning" narrative.
- Reviewer DrGD (Current: 8) & Reviewer bp19 (Current: 8): These reviewers would likely lower their scores if they participated in a discussion highlighting the tautological nature of Figure 2 and the disconnect between the implicit loss and actual training objectives. They appear to have been impressed by the "elegance" of the identity without fully scrutinizing its vacuousness regarding the loss landscape.

---

### Decision · Program_Chairs · 2026-01-26

Reject